# Cyclin-Dependent Kinase 4/6 Inhibitors and Dermatologic Adverse Events: Results from the EADV Task Force “Dermatology for Cancer Patients” International Study

**DOI:** 10.3390/cancers15143658

**Published:** 2023-07-18

**Authors:** Pietro Sollena, Nikolaou Vasiliki, Elias Kotteas, Alexander J. Stratigos, Davide Fattore, Armando Orlandi, Maria Mannino, Marcello Di Pumpo, Monika Fida, Michela Starace, Zoe Apalla, Maria Concetta Romano, Julia Riganti, Sonia Segura, Azael Freites Martinez, Gabriella Fabbrocini, Vincent Sibaud, Ketty Peris

**Affiliations:** 1UOC Dermatologia, Fondazione Policlinico Universitario A. Gemelli IRCCS, 00168 Rome, Italy; pietrosollena@virgilio.it (P.S.); ketty.peris@unicatt.it (K.P.); 2Dermato-Oncology Department, Cutaneous Toxicities Clinic, Andreas Sygros Hospital, National and Kapodistrian University of Athens, 16121 Athens, Greece; drviknik@yahoo.com (N.V.); alstrat2@gmail.com (A.J.S.); 3Oncology Unit, 3rd Department of Medicine, School of Medicine, National and Kapodistrian University of Athens, “Sotiria” General Hospital, 16121 Athens, Greece; ilkotteas@hotmail.com; 4Section of Dermatology, Department of Clinical Medicine and Surgery, University of Naples Federico II, 80126 Naples, Italy; dott.davidefattore@gmail.com (D.F.); gafabbro@unina.it (G.F.); 5Medical Oncology, Fondazione Policlinico Universitario A. Gemelli—IRCCS, 00168 Rome, Italy; armando.orlandi@policlinicogemelli.it; 6Dermatologia, Università Cattolica del Sacro Cuore, 00168 Rome, Italy; mariamannino04@yahoo.it; 7Department of Scienza della Vita e Sanità Pubblica, Università Cattolica del Sacro Cuore, 00168 Rome, Italy; dipumpomarcello@gmail.com; 8Dermatology Service, University Hospital Center “Mother Theressa”, 1005 Tirana, Albania; monikafida@gmail.com; 9Dermatology—IRCCS, Policlinico Sant’Orsola, Department of Specialized, Experimental and Diagnostic Medicine, Alma Mater Studiorum, University of Bologna, 40138 Bologna, Italy; michela.starace2@unibo.it; 10Second Dermatology Department, Medical School, Aristotle University of Thessaloniki, Thessaloniki 54124, Greece; zoimd@yahoo.gr; 11San Camillo Forlanini Hospital, 00152 Rome, Italy; pucciromano55@gmail.com; 12Dermatology Department, Hospital Italiano de Buenos Aires, 1199 Buenos Aires, Argentina; julia.riganti@hospitalitaliano.org.ar; 13Department of Dermatology, Hospital del Mar–Institut Hospital del Mar d’Investigacions Mèdiques (IMIM), Universitat Autònoma de Barcelona (UAB), 08003 Barcelona, Spain; ssegura@psmar.cat; 14Oncodermatology Clinic at Hospital Ruber Juan Bravo and Universidad Europea, 28006 Madrid, Spain; azaelfreites@yahoo.com; 15Oncodermatology Department, Institut Universitaire du Cancer, Toulouse Oncopole, 31500 Toulouse, France; sibaud.vincent@iuct-oncopole.fr

**Keywords:** CDK4/6 inhibitors, skin adverse event, advanced breast cancer

## Abstract

**Simple Summary:**

Treatment with cyclin-dependent kinase 4/6 inhibitor (CDK4/6i), has demonstrated significantly improved progression-free survival in patients with hormone receptor-positive, HER2-negative, advanced breast cancer, when used in combination with endocrine therapies. However, limited data exists on its cutaneous adverse events (AE). The aim of our retrospective study was to investigate the prevalence, types and management of cutaneous AE during CDK4/6i. 79 adult advanced breast cancer patients affected by 125 skin adverse events during treatment with CDK4/6i were recruited at eleven centers. The most frequent cutaneous reactions were pruritus (49/79 patients), alopecia (25/79), and ec-zematous lesions (24/79). We showed that skin reactions are usually mild in severity, and prompt management may limit the negative impact on patients, facilitating beneficial continuation of oncologic treatment.

**Abstract:**

Background: The introduction of cyclin-dependent kinase inhibitors (CDK4/6i) was a great advance in therapeutics for patients with estrogen receptor+/human epidermal growth factor receptor (HER2) locally advanced and metastatic breast cancer. Despite the increasing use of these agents, their adverse drug-related events have not yet been fully characterized. We describe the spectrum of cutaneous adverse reactions occurring in advanced breast cancer patients treated with cyclin-dependent kinase inhibitors, analyzing types, severity, time to onset, and possible treatment outcomes. Methods: We performed a multicentric retrospective study including patients with advanced breast cancer who developed cutaneous lesions during treatment with CDK4/6i in the period from June 2020 to June 2021. Patients > 18 years were recruited at eleven onco-dermatology units located in Albania (1), Argentina (1), France (1), Greece (3), Italy (3), and Spain (2). We evaluated patients’ epidemiological and clinical characteristics, types of cutaneous adverse events, their time to onset, and treatment outcomes. The severity of the skin reactions was assessed using the Common Terminology Criteria for Adverse Events (CTCAE) version 5.0 score. Results: Seventy-nine patients (median age: 62.3 years; range 39–83 years) were included in the study, and, collectively, we recorded a total of 165 cutaneous adverse events during follow-up visits. The most frequent cutaneous reactions were pruritus (49/79 patients), alopecia (25/79), and eczematous lesions (24/79). Cutaneous toxicities were usually mild in severity (>65%) and occurred after a median of 6.5 months. Only four patients (5%) required treatment discontinuation due to the severity of the skin lesions. The majority of the skin reactions were managed with topical treatments. Conclusions: To the best of our knowledge, we present the largest case series of cutaneous adverse events developing in advanced breast cancer patients treated with CDK4/6i. We showed that cutaneous toxicities are usually mild in severity, and manageable with standard supportive care; however, in selected cases, they can lead to treatment discontinuation with possible implications for patients’ clinical outcomes.

## 1. Introduction

Cyclin-dependent kinases (CDK) 4 and 6 are regulatory enzymes that control cell cycle progression from the G1 to the S phase via phosphorylation of several target proteins, causing their activation or inactivation during the G1 phase of the cell cycle [1]. Selective CDK4/6 inhibitors (CDK4/6i) act by blocking the cyclin D1/CDK4/6 complex and inhibit cell cycle transition and thus, cancer cell proliferation as well as endocrine resistance in breast carcinoma [2]. To date, three different orally administrated CDK4/6i have been approved by the European Medical Association (EMA): palbociclib (Ibrance; Pfizer), ribociclib (Kisquali; Novartis), and abemaciclib (Verzenios; Eli Lilly) [3,4,5]. These drugs now represent the standard of care for hormone receptor (HR)-positive, human epidermal growth factor receptor (HER) 2-negative locally advanced and metastatic breast cancer, in combination with endocrine therapy (ET), which represents the largest subgroup of breast cancer. 

The three pivotal trials, PALOMA [3,4], MONALEESA [5], and MONARCH [6], reported significant improvements in progression-free survival versus ET alone (aromatase inhibitors or fulvestrant); nevertheless, a broad spectrum of systemic toxicities has also been reported [3,4,5,6]. 

The three CDK4/6i showed a similar adverse event (AE) profile, with prominent hematologic toxicities such as neutropenia, leukopenia, anemia, and thrombocytopenia [7]. A slightly different toxicity profile was reported for abemaciclib, most likely due to its greater selectivity for CDK4 versus CDK6 [7,8,9]. CDK4 is particularly important for cellular tumorigenesis, while CDK6 is primarily involved in hematopoietic stem cell differentiation [8]. Indeed, abemaciclib showed lower hematologic AE rates compared to palbociclib and ribociclib, but higher gastrointestinal AEs such as diarrhea [8,9]. 

CKD4/6i have been also associated with cutaneous adverse events (cAEs) in pivotal trials [3,4,5,6] and recent literature. The most commonly reported cAEs for the CDK4/6i class were alopecia, skin rash, and pruritus; other common cAEs included eczematous-like reactions [10,11,12,13,14]. The generic term “skin rash” was reported in pivotal studies as a preferred term that equals dermatitis or rash erythematous, maculopapular, palbociclib, or pruritic. In the PALOMA-3 study, palbociclib combined with fulvestrant caused alopecia in 17% of treated patients and cutaneous rash in 15% [3,4]. In the MONALEESA-2 study, ribociclib, combined with letrozole induced alopecia in 33% of patients and cutaneous rash in 18% [4]. In the MONARCH-3 study, abemaciclib plus nonsteroidal AI (anastrozole or letrozole per physician’s choice) caused alopecia in 27% of patients but no skin rash cases were reported [5]. Although less frequent, cases of serious or potentially life-threatening cAEs have been also reported, including bullous dermatitis [10,11,12,13,14]. Finally, we have recently described vitiligo-like reactions with ribociclib and to a lesser degree with palbociclib [15,16]. 

Herein, we characterized the spectrum of cAEs induced by each CDK4/6i in a large cohort of international patients, describing their clinical features, time to onset, treatment approaches and dermatologic outcomes, and their possible impact on an oncologic treatment course. 

## 2. Methods

A retrospective, multicenter cohort study was performed by the European Academy of Dermatology and Venereology (EADV)-Task Force “Dermatology for cancer patients” between June 2020 and June 2021. Adult patients under treatment with CDK4/6i for advanced breast cancer, and who developed dermatologic adverse events during their oncologic therapy, were recruited at eleven onco-dermatology units belonging to: Fondazione Policlinico Universitario A.Gemelli IRCCS, Rome, Italy; Azienda Ospedaliera Universitaria Federico II of Naples, Italy; Policlinico Universitario Sant’Orsola IRCCS, Bologna, Italy; Andreas Sygros Hospital for skin diseases, Athens, Greece; Institut Universitaire du Cancer of Toulouse, France; Aristotle University of Thessaloniki, Thessaloniki, Greece; Papageorgiou Hospital Thessaloniki, Thessaloniki, Greece; Hospital Ruber Juan Bravo and Universidad Europea, Madrid, Spain; Hospital del Mar, Universitat Autònoma de Barcelona, Barcelona, Spain; Hospital Italiano de Buenos Aires, Buenos Aires, Argentina; and the University Medical Center of Tirana “Mother Teresa”, Tirana, Albania. The study was approved by the Institutional Review Board (ID4058), and all patients provided written informed consent; a common IRB approval was obtained, with Fondazione Policlinico Universitario A. Gemelli IRCCS of Rome as the coordinating center.

Patient characteristics were obtained from each medical record, focusing on age, sex, dermatologic medical history, type of CDK4/6i and ET, type and severity of cAEs, time to onset, and therapeutic management of skin toxicities. 

Dermatologists performed cAE diagnosis based on typical clinical manifestations; in uncertain cases, a biopsy was performed for histopathological confirmation. Each cAE was graded according to the Common Terminology Criteria for Adverse Events, version 5.0 (CTCAE v5.0) [17]. Based on our EADV task force consensus and improvement in CTCAE grade, the dermatologic therapy outcome was classified as a complete response (CR), partial response (PR), or no response (NR). A CR was defined as remission of clinical cutaneous lesions and related symptoms; PR was used for at least a 30% decrease in the body surface area (BSA) affected by cutaneous reactions; and NR encompassed lesions with less than a 30% improvement, stable disease, and worsening of the skin AE. 

Multiple skin toxicity types were recorded for CDK4/6i treatment. For each patient, up to three different subsequent cAEs onsets were potentially recorded. For each onset, up to 13 toxicity types were potentially recorded. For each onset and toxicity type, the severity grade (from level 1 to level 3) was recorded.

The distribution of each variable was studied using the Shapiro–Wilk normality test. We reported absolute and relative frequencies for qualitative variables, the mean and standard deviation (SD) for quantitative normally distributed variables, and the median and interquartile range (IQR) for quantitative variables without normal distribution. We performed univariate and multivariate logistic regressions for possible predictors of CDK4/6i-induced cAEs development. In the univariate regression analysis, age, CDK therapy, and pruritus were chosen as predictors for toxicity type and toxicity grade as outcomes for each one of the three onsets. In addition, for each onset, the toxicity grade was not further sub-stratified by toxicity type in the regression analysis.

Relative risk ratios (RRR) were calculated. The statistical significance level was set at *p* < 0.05, and all the analyses were carried out by using the software Stata IC 14 for Mac (Stata Corp, Lakeway, TX, USA).

## 3. Results

Seventy-nine patients were enrolled in the study, and their clinical and epidemiological characteristics are shown in Table 1. The median age was 62.3 years (range 39–83 years), 78 patients (98.7%) were females, and 1 (1.3%) was male; all patients were diagnosed with HR+/HER2− metastatic breast cancer. In total, 24 patients were on treatment with palbociclib (30.4%), 54 with ribociclib (68.3%), and 1 with abemaciclib (1.3%). 

Collectively, we reported 165 cAEs (Table 1): pruritus was the most common toxicity (*n =* 49, 62%), followed by alopecia (*n =* 25, 32%), and eczematous rash (*n =* 24, 31%). The median time to onset of cAEs (range 1–18 months) was 6 months for patients treated with palbociclib, 6.5 months for ribociclib, and 9 months for abemaciclib. Most cAEs (86%) were mild in severity and categorized as grade 1 or 2 (*n* = 158); only seven cases were graded as moderate to severe (grade 3). A total of 36 patients reported more than one cAE: 22 of them presented with three different cAEs, and 3 reported four different cAEs at the time of the dermatologic consultation. The causality was usually assessed thanks to our dermatological task force’s expertise and the temporal relationship between the development of cAE and the start of CDK4/6i; in case of doubt or confounding factors, the “Adverse Drug Reaction Probability Scale” developed by Naranjo was performed (Appendix A).

All patients affected by a cAE during CDK4/6i treatment were managed with skin-directed or systemic dermatologic therapy, with a complete or partial response in 65/79 cases (Table 1). Topical high-potency steroids alone, or in combination with topical moisturizers, were the preferred treatment choice (*n =* 48, 61%). Systemic dermatologic therapy was required in a subset of patients with moderate to severe reactions, including oral prednisone in seven patients, UVA/UVB phototherapy in six patients, doxycycline 100 mg daily in five patients, oral antihistamines in three patients, and spironolactone 200 mg daily in two patients.

The majority (> 90%) of patients did not discontinue CDK4/6i therapy due to their cAEs; however, seven patients underwent oncologic treatment modification (temporary interruption and dosage change), with an improvement of their skin toxicities, and four patients experienced permanent drug discontinuation due to their non-response despite skin-directed therapy implementations. None developed grade 4 reactions, requiring hospitalization or intensive care support.

## 4. Toxicity Subtypes

### 4.1. Pruritus

An itchy sensation was reported by 49 patients (62%): 37 of these were on treatment with ribociclib, 11 with palbociclib, and 1 with abemaciclib. The average time to onset was 3 months from CDK4/6i administration; this was an earlier cAE compared to the other skin toxicities (Figure 1).

Pruritus also occurred in association with other cAEs: eczematous rash (*n =* 18), vitiligo-like lesions (*n =* 16), and maculopapular rash (*n =* 14).

Treatment included topical emollients twice daily and oral antihistamines for mild cases, while topical and/or oral corticosteroids were prescribed for the most severe cases. In two patients, the pruritic rash was classified as CTCAE grade 3, requiring oral steroids. None of our patients underwent permanent CDK4/6i withdrawal due to pruritus; however, one patient required two weeks of drug course suspension due to the development of pruritic lesions, which resolved after 14 days of topical high-potency steroids and systemic antihistamines. Treatment outcomes reported PR in 29/49 cases and CR in 20/49 cases.

### 4.2. Eczematous Lesions

We reported an eczema-like rash in 24/79 patients who were on treatment with ribociclib (*n =* 16) and palbociclib (*n =* 8). The time to onset was between 6 and 9 months from CDK4/6i treatment initiation. 

Eczematous reactions consisted of multiple erythematous scaly papules, sometimes arising as localized patches/plaques or dyshidrotic vesicles, often localized on the posterior trunk (Figure 2).

In 75% (*n =* 18) of cases, the rash was accompanied by pruritus. Lesions were often mild in severity (15/24 CTCAE grade 1; 8/24 grade 2; and 1/24 grade 3); however, three patients required a dose modification of the oncologic drug, and one patient underwent permanent CDK4/6i treatment discontinuation. 

All patients with eczematous rash reported a response to treatment, with PR in 13/24 cases and CR in 11/24. Therapeutic approaches included topical corticosteroids or a combination of topical steroids and antihistamines or antibiotics. Seven patients also required the addition of systemic treatment: four with oral steroids, two with antihistamines, and one with antibiotics. 

### 4.3. Maculopapular Reaction

Seventeen patients (22%) developed maculopapular rashes (MPR), most often under ribociclib treatment (11/17). The average time to onset was slightly earlier with palbociclib than with ribociclib (4.5 months and 7 months, respectively).

The clinical presentation was characterized by nonspecific morbilliform, erythematous, round lesions (Figure 2) that were mild in severity (11/17 CTCAE grade 1; 6/17 grade 2). MPR occurred mainly on the trunk and the upper limbs; the lesions were usually itchy (14/17), although they could also develop asymptomatically.

Treatment of MPR included emollients and potent topical steroid therapy, achieving a CR in the majority of patients (CR 13/17; PR 4/17). 

### 4.4. Alopecia

Alopecia was reported in 25/79 patients, regardless of the CDK4/6i (15 patients on ribociclib, 9 on palbociclib, and 1 on abemaciclib). The median time to onset was 8.2 months (range 1–18) from CDK4/6i therapy initiation, with no significant differences among the CDK4/6i. In all cases, dermatologists clinically identified it as androgenetic alopecia, with increased hair loss in the central area of the scalp (Figure 2). Alopecia was mild in severity in the majority of the cases (*n =* 21, 84% grade 1). Due to the major clinical pattern being androgenetic alopecia-like, the assessment was evaluated using the Ludwig scale together with the evaluation of the hair loss association with the pull test. The Ludwig scale is based on three degrees (grades I, II, III), and they reported the first degree. No Response was considered no Improvement of the scale degree with a positive pull test; Partial Response was for improvement more of one grade on the same scale with a negative pull test; and Complete Response was when the patient achieved the remission of the occurred alopecia with a negative pull test. No discontinuations were due to alopecia.

Treatment approaches consisted of minoxidil 5% topical solution alone (11/25) or in combination with spironolactone 200 mg, daily (2/25). Concerning treatment outcomes, we reported a PR in 15/25 cases, CR in 3/25, and NR in 7/25.

### 4.5. Vitiligo-like Lesions

Twenty-one patients presented with vitiligo-like lesions (VLLs): *n =* 1 (5%) were on treatment with palbociclib and *n =* 20 (95%) with ribociclib. Time to onset was slightly later than other cAEs, with a median of 9.8 months (range 4–16). VLLs were non-segmental/bilateral in all 21 patients; 14/21 had a localized form, mainly distributed on the chest and/or arms, and 7/21 had the generalized subtype. The cutaneous lesions consisted of hypopigmented macules and patches with poorly defined borders (Figure 2). Clinical grading was mild in the majority of the cases (*n =* 13, 62% grade 1; *n =* 6, 28% grade 2) and only two cases were grade 3 (10%).

VLLs were often associated with pruritus (16/21), which sometimes anticipated the onset of hypopigmented lesions. Medium-high potency topical corticosteroids were the treatment of choice in all 21 patients, in combination with topical calcineurin inhibitors in 5 patients and with UVB phototherapy in 6 patients. During oncologic treatment, 12 patients achieved a PR (improvement > 30% BSA involved) limited to facial lesions, while 9 patients did not experience any therapeutic improvement.

### 4.6. Lichenoid Reactions

We assessed lichenoid reactions in seven patients, often in association with pruritus (5/7 cases). Lichenoid lesions affected patients treated with all types of CDK4/6i (four ribociclib, two palbociclib, and one abemaciclib).

Toxicities were ranked as CTCAE grade 1 in almost all cases (6/7), and only one patient only presented with CTCAE grade 2 toxicity. The median time to onset was 7.4 months from treatment initiation, and it was delayed in comparison to pruritus and maculopapular rashes. The clinical presentation was heterogeneous, ranging from typical lichen planus with papules and visible Wickham striae to hypertrophic or squamous lesions. Lichenoid reactions mainly developed on the trunk and the limbs.

Treatment was based on topical steroids and moisturizers, and in one case, a topical and systemic steroid combination was required to achieve a CR (CR 1/9; PR 8/9). 

### 4.7. Other Less Common cAEs to CDK4/6i

We assessed other less common skin toxicities including papulo-pustular rashes (5/79), psoriasis (4/79), asteatotic skin (4/79), cutaneous lupus (2/79), nail dystrophy (3/79), hand and feet reactions (2/79), acral pigmentation (1/79), and localized morphea (1/79). 

Papulo-pustular rashes typically manifested as multiple small follicular papules located on the face and trunk, particularly on the chest; the bacterial swabs were negative for aseptic dermatitis. Psoriatic lesions were all vulgaris in type, with erythematous and scaly plaques mainly located on elbows, knees, and the lower back. Two of the four patients had a personal history of psoriasis, which worsened after CDK4/6i treatment. Two patients experienced subacute cutaneous lupus erythematosus (SCLE), with the development of erythematous and infiltrated scaly plaques on the face, chest, and forearms; ANA and Ro/SSA antibodies were positive in both cases. The first patient had a previous history of SCLE, which flared up after four months of palbociclib therapy. The second patient turned ANA negative after two months of ribociclib withdrawal and then re-started the same CDK4/6i (ribociclib) without clinical relapses on follow-up visits. Nail disorders were not routinely recorded and they were only documented when reported by the patient (3/79).

### 4.8. Clinical Predictors of Cutaneous Toxicities

According to our univariate logistic regression analysis, age was a significant predictor of VLL as a second onset cAE (*p* = 0.043, RRR = 1.16). Age was also a significant predictor of grade 2 severity in cases of first onset (*p* = 0.018, RRR = 0.93) and second onset skin reactions (*p* = 0.089, RRR = 1.29), respectively. Pruritus was found to be a significant predictor of eczematous rash, (*p* = 0.001, RRR = 14.00), maculopapular rash (*p* = 0.011, RRR = 11.20), lichenoid, (*p* = 0.050, RRR = 11.20), and a vitiligo-like reaction (*p* = 0.011, RRR = 6.72).

The multinomial logistic regression analysis reported a non-significant figure for age, type of CDK4/6i therapy, and pruritus in predicting the type of first onset cAE (*p* = 0.052; pseudo R^2^ = 0.12). Conversely, the multinomial logistic regression showed age, type of CDK4/6i therapy, and pruritus to be significant predictors of severity of the first onset cAE (*p* = 0.016; pseudo R^2^ = 0.15). There was no significant association between the type of CDK4/6i and the specific type of cAE reported.

## 5. Discussion

We reported the experience of the EADV task force “dermatology for cancer patients” who examined the largest case series regarding the spectrum of skin toxicities recorded in breast cancer patients undergoing treatment with CDK4/6i, including each of the approved and available CDK4/6i. CAEs can be an important issue and may directly impact the oncologic treatment outcome, patients’ compliance, and their quality of life [12]. In our cohort, we found pruritus as the earliest and most frequent cAE, often developing concomitantly with eczematous or maculopapular reaction, or preceding cutaneous lesions. The percentage of patients affected by pruritus (62%) was higher in comparison with previously reported literature data (~18%); however, the lower figure in the literature may stem from the underreporting of the cutaneous symptoms as well as the lack of systematic dermatologic consultations in previous studies [11,12,13,14]. Moreover, it Is well known that pruritus could arise as the first symptom of other incoming cutaneous toxicity (e.g., lichenoid, eczematous, and bullous). Due to the established expertise of our taskforce dermatologists, it may be that our patients were sent for a dermatological examination earlier, at the onset of the first itchy symptoms. This could also explain the absence of bullous reactions in our study group, a result different from previous studies [13,14], which was probably due to earlier dermatologic therapeutic intervention. Pruritus occurred regardless of CDK4/6i type, and it seemed to be a class-related cAE; however, the specific underlying pathogenetic mechanisms remain unclear. Itchy skin in the context of CDK4/6i is a therapeutic challenge since pruritus is often resistant even to high doses of oral antihistamines; high-potency topical and/or systemic steroids in addition to emollient creams are often needed to mitigate the symptoms. Although pruritus was a very frequent cAE, it seldom led to permanent interruption of CDK4/6i; nevertheless, one of our patients required a period of drug interruption to achieve CR.

An eczematous reaction was the most frequently reported cAE in our cohort. Eczematous lesions can be often misdiagnosed as maculopapular or lichenoid rashes by non-dermatologist physicians; thus, a diagnostic biopsy should be indicated in any case of clinical doubt. This is further supported by the heterogeneous clinical presentation of lichenoid and maculopapular lesions, often depending on the association with pruritus. Indeed, the maculopapular rash reported in our patient cohort was also frequently associated with an itching sensation, but the skin lesions were less crusty and more erythematous, and slightly raised on the skin surface (Figure 2). Maculopapular reactions, such as itching, may also precede the onset of other skin diseases, and, in agreement with previous studies [13,14], it occurred early in our patients (Figure 1).

In line with the existing literature data [12,13,14], we found alopecia as a very frequently reported cAE, probably due to the synergetic effect between CDK4/6i and ET which are prescribed in combination for breast cancer treatment. Indeed, it has been already reported that the risk of alopecia increases in patients receiving combined CDK4/6i and ET treatment compared with those treated with ET alone [18]. The management of alopecia is challenging, although patients seldom require specific treatments. Due to the probable hormonal pathogenesis, we suggest topical minoxidil 5% treatment and adding systemic spironolactone (200 mg tablet) in more resistant cases.

The occurrence of VLLs is not infrequent during CDK4/6i therapy, and it has been already reported in the literature [15,16,19]. Our study cohort showed a significantly higher proportion of VLLs with ribociclib compared to palbociclib; the discrepancy has not yet been clarified in the literature, but the activity and toxicity profiles of the two drugs seem to account for this phenomenon. Despite belonging to the same class of drugs, ribociclib and palbociclib inhibit CDK4 and CDK6 as well as other kinases in very different ways [20]. In addition to justifying a difference in efficacy (ribociclib increases the overall survival in all pivotal studies, whereas palbociclib has no significant impact on overall survival), this has an impact on the profile of different side effects, such as liver toxicity and QTc increase, which occur significantly more frequently during ribociclib treatment [4,5]. Moreover, the liver toxicity of ribociclib appears to be immune-mediated, and the damage to and dysregulation of the immune system can be the cause of vitiligo [19]. Taking into account these factors, the almost exclusive occurrence of VLLs during treatment with ribociclib as compared to palbociclib may be attributable to a distinct hepatic and skin toxicity profile, as well as an autoimmune response stimulation linked to the different kinoma of the two drugs. Interestingly, VLLs developed later than other cAEs (> 9 months), and they were often preceded or accompanied by pruritus. Despite limited clinical efficacy, treatment is based on topical steroids with or without the association of topical calcineurin inhibitors; we suggest using systemic treatment with UVB-narrow band phototherapy in more resistant cases.

Herein, we confirm SCLE could be a possible cAE during CDK4/6i treatment, as already reported in the literature [13,14]. Two of our patients experienced SCLE but interestingly, we reported the first ribociclib-induced case since only SCLE cases developed during palbociclib treatment have previously been described [13,14]. Cutaneous lesions consisted of erythematous and infiltrated scaly plaques mainly distributed on sun-exposed areas (face, chest, and forearms), similar to other cases previously reported [14].

In our cohort of patients, we reported several auto-inflammatory cAEs, some of which are usually referred to as immune-related cutaneous adverse events to immune checkpoint inhibitors (ICI), such as anti-PD1 and anti-PDL-1 agents [20,21]. Recent studies uncovered novel therapeutic potentials for CDK4/6i, suggesting an immunomodulatory effect that goes beyond the intrinsic anti-tumor properties related to cell cycle inhibition [22]. It is known that CDK4/6i has direct effects on T lymphocytes, with the reduction in T regulatory cells and direct activation of effector T lymphocytes leading to a stronger anti-tumor immune response [23,24]. In addition, CDK4/6i have been shown to increase tumor cell antigen presentation via upregulation of the major histocompatibility complex (MHC) class I, thereby enhancing cancer cell immunogenicity and recognition by the immune system [25,26]. Altogether, this evidence poses the rationale for the development of a toxicity profile that encompasses the spectrum of immune-related cAEs, an example being VLLs which have been also reported in our patient population. 

The spectrum of CDK4/6i Induced cAEs shares some similarities with the dermatological toxicities caused by ET for adjuvant breast cancer treatment. ET encompasses aromatase inhibitors (AI) and selective estrogen receptor modulators (SERM), and the most commonly reported cAEs include pruritus, alopecia, a clinically heterogeneous cutaneous rash, skin flushing, vulvovaginal atrophy, and connective tissue disorders [27,28,29]. A meta-analysis on ET-induced alopecia reported a hair loss incidence of 2.2–2.5% and 9% for breast cancer patients on adjuvant AI and SERM, respectively [30]. Both female pattern hair loss and male pattern androgenetic alopecia have been clinically described, and they are usually responsive to 5% topical minoxidil therapy [31]. Conversely, a minority of patients may suffer from low-grade ET-induced hirsutism, which is easily manageable with topical interventions such as epilation and laser therapy [32]. Skin flushing and vulvovaginal atrophy are also linked to declining estrogen levels, with dysfunctional hypothalamic temperature center regulation, decreased blood flow, and dermal collagen and elastin thinning of the vulvovaginal region. Connective tissue disorders have been seldom reported, including cutaneous vasculitis, subacute cutaneous lupus erythematosus, and erythema nodosum, with variable response to corticosteroids and clinical improvement upon ET discontinuation [33,34]. 

Treatment approaches of CDK4/6i-induced cAEs included high- to very-high-potency topical steroids (e.g., betamethasone or clobetasol propionate) and/or topical moisturizers (Figure 3). The use of systemic corticosteroids (prednisone 0.5–1 mg/kg/day), oral antihistamines, and UVB phototherapy was restricted to persistent and severe reactions (i.e., CTCAE grade 2 or 3). 

Despite the occurrence of cAEs, almost all our patients (95%) continued CDK4/6i therapy. We suggest, unless otherwise recommended by a dermatologist, to continue CDK4/6i treatment in cases of grade 1 and 2 cAEs, and to interrupt it in cases of grade 3 cAEs or more. Upon improvement of a grade ≥ 3 cAE to grade ≤ 1, a re-challenge with a reduced dose of CDK4/6i should be considered. 

With regard to the inferential analysis, significant findings were scarce, as shown in the Results section. We did not find a relevant difference between the specific CDK4/6i and types of cAEs, probably due to the relatively small number of patients included or to the class specificity of cutaneous reactions. The main significance was found for the multinomial logistic regression model for severity of the first toxicity as outcome and age, CDK4/6i, and pruritus as predictors. However, the model showed a low pseudo R^2^ (0.15). The results are preliminary, and this could be overcome in the future as new data become available for analysis. 

## 6. Conclusions

To the best of our knowledge, we reported the biggest case series of cAEs occurring in breast cancer patients treated with all the CDK4/6i treatments available. This paper highlights the possibility of continuing CDK4/6i treatment despite skin reactions in a large majority of patients exposed to cutaneous toxicities. It is still challenging to establish the optimal cAE treatment, to decide on a short interruption of the oncologic therapy, or to switch to a different CDK4/6i. Prevention, early recognition, and adequate intervention are required for maintaining the right dose and mitigating cAE severity. 

## Figures and Tables

**Figure 1 cancers-15-03658-f001:**
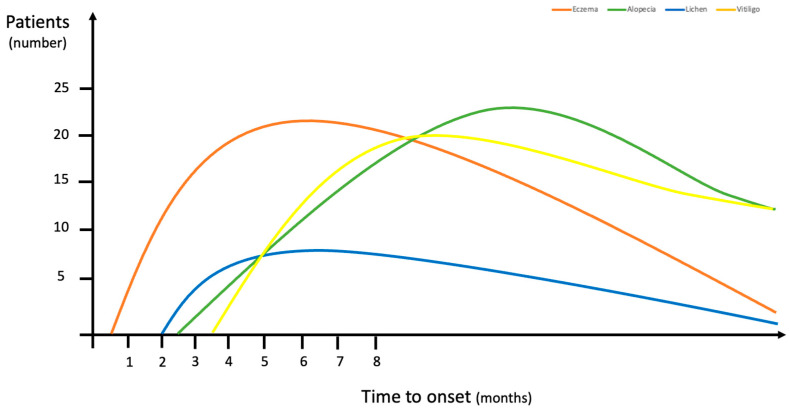
Time of onset for cutaneous adverse events since starting treatment with CDK4/6 inhibitors.

**Figure 2 cancers-15-03658-f002:**
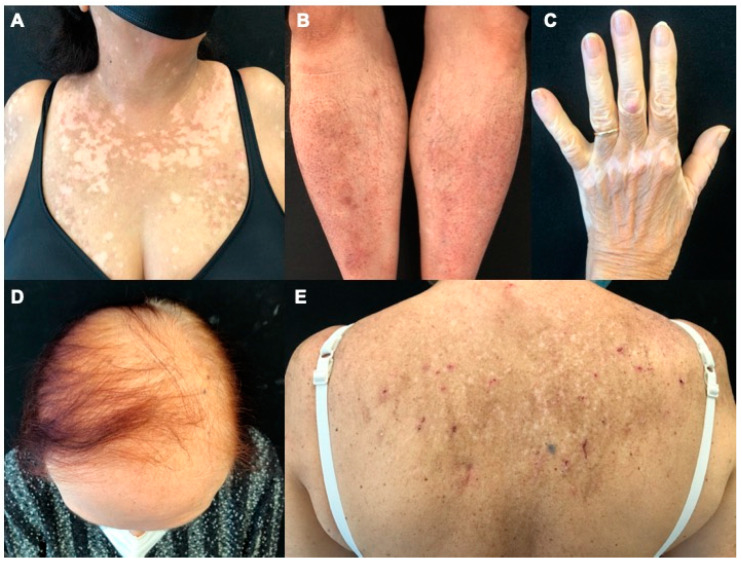
Clinical pictures of cutaneous adverse events to CDK4/6i. (**A**), Vitiligo-like lesions made of multiple hypopigmented macules with ill-defined margins on the patient’s sun-exposed chest. (**B**), Erythematous maculopapular lesions on bilateral pretibial areas. (**C**), Vitiligo-like lesions located on the back of the left hand. (**D**), Alopecia of the central area of the scalp. (**E**), Eczematous reaction on patient’s back showing few scratched maculopapular lesions on the post-inflammatory hyperpigmented area.

**Figure 3 cancers-15-03658-f003:**
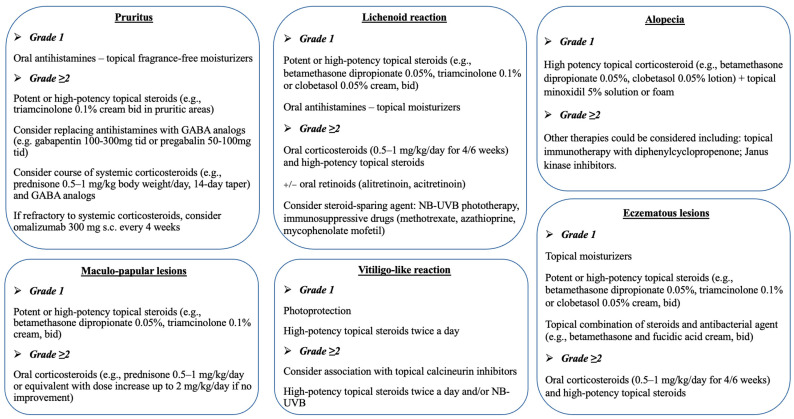
Management of cutaneous adverse events during treatment with CDK4/6 inhibitors.

**Table 1 cancers-15-03658-t001:** Patients’ clinical and epidemiological characteristics.

Patient No./Sex/Age	Oncologic Therapy	Type of cAE, Reaction1/Reaction2/Reaction3/Reaction4	Months of CDK4/6i Treatment at cAE Diagnosis, Reaction1/2/3/4, No.	Treatment cAE 1	Treatment cAE 2	3	4	Systemic Treatment	Outcome cAE1	Outcome cAE2	Outcome cAE3	Outcome cAE 4
1/F/63	Palbociclib	Pruritus/maculopapular rash/vitiligo	8/10/12	high potency steroid cream	high potency steroid cream	high potency steroid cream		none	PR	PR	NR	
2/F/79	Ribociclib+letrozole	Pruritus/vitiligo	6/16	high potency steroid cream	high potency steroid cream+calcineurin inhibitors			none	PR	PR		
3/F/66	Ribociclib+fulvestrant	Vitiligo	6	high potency steroid cream+calcineurin inhibitors				UVA/UVB phototherapy	PR			
4/F/54	Ribociclib+fulvestrant	Pruritus/vitiligo/maculopapular rash	8/8/9	high potency steroid cream	high potency steroid cream	high potency steroid cream		UVA/UVB phototherapy	CR	PR	CR	
5/F/48	Ribociclib+letrozole	Pruritus/maculopapular rash /vitiligo	4/4/6	high potency steroid cream+emollient cream	high potency steroid cream	high potency steroid cream+calcineurin inhibitors		none	PR	CR	NR	
6/F/73	Ribociclib+letrozole	Pruritus/maculopapular rash /vitiligo	4/8/16	high potency steroid cream+antihistamine cream	high potency steroid cream	high potency steroid cream+calcineurin inhibitors		none	PR	CR	NR	
7/F/59	Ribociclib+letrozole	Pruritus/maculopapular rash /vitiligo	6/10/12	high potency steroid cream	high potency steroid cream	high potency steroid cream		none	PR	PR	PR	
8/F/56	Ribociclib+letrozole	Pruritus/eczematous rash/alopecia	4/5/7	high potency steroid cream	high potency steroid cream	minoxidil		steroid	PR	PR	PR	
9/F/50	Ribociclib+letrozole	Pruritus/asteatosis/eczematous rash	5/6/8	high potency steroid cream	high potency steroid cream+emollient cream	high potency steroid cream+emollient cream		none	CR	CR	PR	
10/F/68	Ribociclib+letrozole	Pruritus/asteatosis/vitiligo	4/7/9	high potency steroid cream+antihistamine cream	high potency steroid cream	high potency steroid cream+calcineurin inhibitors		none	CR	CR	PR	
11/F/64	Ribociclib+letrozole	Pruritus/Papulo-pustular rash	7/8	combo steroid+antiseptic/antibiotic cream	combo steroid+antiseptic/antibiotic cream			antibiotic	CR	CR		
12/F/82	Ribociclib+letrozole	Pruritus/eczematous rash	8/12	high potency steroid cream	high potency steroid cream			none	PR	PR		
13/F/63	Ribociclib+letrozole	Pruritus/alopecia/maculopapular rash	6/9/11	high potency steroid cream	high potency steroid cream	high potency steroid cream+emollient cream		none	PR	CR	CR	
14/F/63	Ribociclib+letrozole	Pruritus/eczematous rash /vitiligo	5/6/10	high potency steroid cream	high potency steroid cream	high potency steroid cream+calcineurin inhibitors		none	PR	PR	NR	
15/F/68	Ribociclib+fulvestrant	Pruritus/Lichenoid reaction	5/7	high potency steroid cream	high potency steroid cream			steroid	CR	CR		
16/F/69	Ribociclib+letrozole	alopecia	6	minoxidil				none	PR			
17/F/65	Ribociclib+letrozole	alopecia	11	minoxidil				none	PR			
18/F/59	Abemaciclib	Pruritus/Lichenoid reaction/alopecia	5/9/12	high potency steroid cream	high potency steroid cream	minoxidil		none	PR	PR	NR	
19/F/61	Ribociclib+letrozole	Pruritus/ maculopapular rash	6/8	high potency steroid cream	high potency steroid cream			none	PR	CR		
20/F/56	Ribociclib+letrozole	Nail dystrophy/alopecia	5/8	combo steroid+antiseptic/antibiotic cream	minoxidil			antibiotic	PR	PR		
21/F/42	Ribociclib+letrozole	Pruritus/ maculopapular rash	4/4	high potency steroid cream	high potency steroid cream			none	CR	CR		
22/F/77	Ribociclib+letrozole	Pruritus/ maculopapular rash	7/8	high potency steroid cream	high potency steroid cream			steroid	CR	CR		
23/F/50	Ribociclib+letrozole	alopecia	5	minoxidil				none	PR			
24/F/61	Ribociclib+anastrazole	Pruritus/eczematous rash	8/10	high potency steroid cream	high potency steroid cream			none	PR	PR		
25/F/76	Ribociclib+letrozole	Pruritus/alopecia	4/8	high potency steroid cream	high potency steroid cream+minoxidil			none	PR	NR		
26/F/72	Ribociclib+letrozole	Pruritus/Cutaneous lupus	3/5	high potency steroid cream+emollient cream	high potency steroid cream+emollient cream			none	CR	PR		
27/F/80	Ribociclib+letrozole	Pruritus/eczematous rash/Hand and Foot reaction	¾/14	high potency steroid cream+emollient cream	high potency steroid cream+emollient cream	combo steroid+antiseptic/antibiotic cream		steroid	CR	CR	PR	
28/F/56	Palbociclib+letrozole	Lichenoid reaction	12	emollient cream				none	PR			
29/F/44	Palbociclib+letrozole	Pruritus/eczematous rash/papulopustular rash	7/14/17	high potency steroid cream	high potency steroid cream+metronidazole cream	combo steroid+antiseptic/antibiotic cream		antibiotic	PR	PR	CR	
30/M/46	Ribociclib	Papulo-papulopustular rash	1	combo steroid+antiseptic/antibiotic cream				antibiotic	PR	PR		
31/F/42	Ribociclib+letrozole	Pruritus/vitiligo	4/14	mild steroid cream	mild steroid cream			UVA/UVB phototherapy	PR	NR		
32/F/45	Palbociclib+letrozole	Cutaneous lupus	4	mild steroid cream	mild steroid cream+calcineurin inhibitors			none	PR			
33/F/60	Ribociclib+letrozole	Pruritus/eczematous rash/alopecia	4/7/10	high potency steroid cream	high potency steroid cream	minoxidil		antihistamine	PR	PR	PR	
34/F/60	Ribociclib+letrozole	eczematous rash/vitiligo	8	mild steroid cream	high potency steroid cream			none	CR	PR		
35/F/64	Ribociclib+fulvestrant	papulopustular rash/eczematous rash	2/8	mild steroid cream	mild steroid cream			steroid + antibiotic	CR	CR		
36/F/67	Ribociclib+letrozole	Pruritus/asteatosis	5/8	high potency steroid cream	emollient cream			none	PR	PR		
37/F/63	Ribociclib+letrozole	Vitiligo	10	high potency steroid cream+calcineurin inhibitors				none	NR			
38/F/68	Ribociclib+letrozole	Pruritus/papulopustular rash /eczematous rash/vitiligo	4/5/10/13	high potency steroid cream	combo steroid+antiseptic/antibiotic cream	high potency steroid cream	high potency steroid cream	steroid	PR	PR	PR	PR
39/F/72	Ribociclib+fulvestrant	Pruritus/eczematous rash	5/15	high potency steroid cream	high potency steroid cream			steroid	CR	CR		
40/F/39	Ribociclib+letrozole	vitiligo	11	high potency steroid cream				UVA/UVB phototherapy	PR			
41/F/46	Ribociclib+letrozole	alopecia	8	none				none	NR			
42/F/78	Palbociclib+fulvestrant	alopecia	9	none				none	NR			
43/F/70	Palbociclib+letrozole	Alopecia/eczematous rash	10/12	mild steroid cream	mild steroid cream	high potency steroid cream		none	PR	CR		
44/F/74	Ribociclib+letrozole	Pruritus/psoriasis/alopecia/Vitiligo	1/5/8/10	high potency steroid cream	high potency steroid cream+combo steroid+antiseptic/antibiotic cream	minoxidil	high potency steroid cream+calcineurin inhibitors	none	PR	PR	CR	NR
45/F/71	Palbociclib+letrozole	alopecia	6	none				none	NR			
46/F/76	Palbociclib+letrozole	alopecia	7	none				none	NR			
47/F/68	Palbociclib+letrozole	alopecia	5	none				none	NR			
48/F/70	Ribociclib+letrozole	eczematous rash/alopecia	12/14	mild steroid cream	minoxidil			none	PR	PR		
49/F/73	Palbociclib+fulvestrant	alopecia	7	minoxidil				none	PR			
50/F/83	Ribociclib+letrozole	Pruritus/eczematous rash/Vitiligo	¾/8	high potency steroid cream	combo steroid+antiseptic/antibiotic cream	high potency steroid cream		antihistamine	PR	PR	NR	
51/F/54	Palbociclib+letrozole	Alopecia/eczematous rash	8/10	minoxidil	high potency steroid cream			spironolactone	PR	CR		
52/F/61	Palbociclib+letrozole	Alopecia	10	minoxidil				none	CR			
53/F/52	Ribociclib+letrozole	Eczematous rash/Alopecia	14/18	high potency steroid cream	minoxidil			none	PR	PR		
54/F/58	Palbociclib+letrozole	Pruritus/eczematous rash	2/4	high potency steroid cream	high potency steroid cream			none	CR	CR		
55/F/63	Palbociclib+letrozole	Pruritus/eczematous rash	2/3	high potency steroid cream	high potency steroid cream			none	CR	CR		
56/F/59	Palbociclib+fulvestrant	Pruritus/ maculopapular rash	3/6	high potency steroid cream	high potency steroid cream			none	CR	CR		
57/F/58	Palbociclib+letrozole	Pruritus/ maculopapular rash	4/5	high potency steroid cream	high potency steroid cream			none	CR	CR		
58/F/51	Palbociclib+letrozole	Pruritus/alopecia/Psoriasis	5/6/10	mild steroid cream	minoxidil	mild steroid cream		none	PR	PR	PR	
59/F/64	Ribociclib+letrozole	Alopecia	7	minoxidil				spironolactone	PR			
60/F/67	Palbociclib+letrozole	Pruritus/eczematous rash	4/4	high potency steroid cream	high potency steroid cream			none	CR	CR		
61/F/59	Palbociclib+letrozole	Maculopapular rash	2	high potency steroid cream				none	CR			
62/F/68	Palbociclib+letrozole	Pruritus/eczematous rash	2/3	high potency steroid cream	high potency steroid cream			none	CR	CR		
63/F/62	Ribociclib+letrozole	Pruritus/maculopapular rash	4/4	high potency steroid cream	high potency steroid cream			none	CR	CR		
64/F/56	Ribociclib+letrozole	Pruritus/psoriasis	4/6	high potency steroid cream+emollient cream	high potency steroid cream+emollient cream			none	PR	PR		
65/F/62	Ribociclib+anastrazole	Pruritus/Alopecia/eczematous rash	5/6/8	mild steroid cream	minoxidil	mild steroid cream+emollient cream		none	PR	PR	PR	
66/F/46	Ribociclib	Maculopapular rash/Alopecia	5/8	mild steroid cream	minoxidil			none	PR	PR		
67/F/63	Palbociclib+fulvestrant	Pruritus/Lichenoid reaction	6/8	high potency steroid cream	high potency steroid cream			none	PR	PR		
68/F/80	Ribociclib+letrozole	Vitiligo/maculopapular rash	10/10	high potency steroid cream+calcineurin inhibitors	high potency steroid cream			UVA/UVB phototherapy	PR	PR		
69/F/58	Ribociclib+letrozole	Pruritus/Vitiligo	5/15	high potency steroid cream	high potency steroid cream+calcineurin inhibitors			UVA/UVB phototherapy	PR	PR		
70/F/64	Palbociclib+fulvestrant	Pruritus/eczematous rash	7/9	calcineurin inhibitors	high potency steroid cream			none	CR	CR		
71/F/69	Ribociclib+letrozole	lichenoid reaction/vitiligo	8	high potency steroid cream	high potency steroid cream			none	PR	PR		
72/F/53	Ribociclib+letrozole	Pruritus/Vitiligo	4/8/9	high potency steroid cream	high potency steroid cream+calcineurin inhibitors			antihistamine	PR	PR		
73/F/71	Ribociclib+letrozole	Pruritus/lichenoid reaction /Vitiligo	6/8/12	high potency steroid cream	high potency steroid cream	high potency steroid cream+calcineurin inhibitors		none	PR	PR	NR	
74/F/55	Ribociclib+letrozole	Pruritus/ maculopapular rash	6/8	emollient cream	emollient cream			none	CR	CR		
75/F/49	Ribociclib+letrozole	Pruritus/lichenoid reaction	4/7	high potency steroid cream	high potency steroid cream			none	PR	PR		
76/F/66	Palbociclib+letrozole	Asteatosis/Nail dystrophy	2/6	emollient cream	emollient cream+calcineurin inhibitors			none	CR	PR		
77/F/68	Palbociclib+fulvestrant	morphea	3	high potency steroid cream+calcineurin inhibitors				none	PR			
78/F/82	Ribociclib+fulvestrant	Pruritus/maculopapular rash/Hand and Foot reaction	2/4/6	high potency steroid cream+antihistamine cream	high potency steroid cream+emollient cream	high potency steroid cream+emollient cream		none	CR	CR	PR	
79/F/47	Ribociclib+examestane	Pruritus/eczematous rash/ acral hyperpigmentation/Nail dystrophy	¼/8/12	high potency steroid cream+antihistamine cream	high potency steroid cream+emollient cream	high potency steroid cream+emollient cream	high potency steroid cream+calcineurin inhibitors	none	PR	PR	PR	PR

## Data Availability

The data presented in this study are available on request from the corresponding author.

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
