# Peer review of "Cyclin-Dependent Kinase 4/6 Inhibitors and Dermatologic Adverse Events: Results from the EADV Task Force “Dermatology for Cancer Patients” International Study"

_cancers, 2023, doi:10.3390/cancers15143658_

Round 1

Reviewer 1 Report

Overall well written paper of a retrospective multicentric cohort and includes palbociclib, ribociclib and abemaciclib.

Paper provides minimal new information compared to other papers. The main strength of the cohort is that multiple drugs are included. The authors should expand on this further. More data on comparisons between the 3 drugs- to illustrate as to whether they are any nuances between them.

Could be improved by citing other articles look at large cohorts of C4/C6 inhibitors and skin toxicities and drawing comparisons from these.

It would be interesting to know to total numbers of patients treated to get an idea of incidence of skin toxicity.

Reviewer 2 Report

In the Introduction- Please describe the cutaneous toxicities as reported in pivotal trials in greater detail as that is the main focus of the paper

Methods: 

1: Please clarify if a common IRB approval was obtained for all sites or separate for each site

2: Methods:

a) How were definitions of response criteria agreed upon? Is there a reference for this or were they consensus based? 

b) Also specify treatment guidelines followed. When were only topical measures and when systemic therapy was initiated? I think this is shown in figure 1 but would be good to mention in methods as it prepares the reader as to how these toxicities were managed and what guidelines were followed

Results:

a) How was response in alopecia assessed? were there any discontinuations due to alopecia?

b) How was causality of these cutaneous reactions with CDK4 inhibitors assessed? Did the patients who require dose modification or discontinuation have improvement in their toxicities with such an approach? That needs to be specified as there may be a lot of confounding factors

c) It is very hard to follow the regression analysis without any tables. would suggest removing it and keeping this descriptive or providing more details and explaining it better

Discussion:

a) There was a significantly higher proportion of VLL with ribociclib compared to palbociclib- that was not discussed much- Any hypothesis/data on that in literature?

b) I would suggest shortening paragraph 5 of discussion discussing auto-inflammatory side effects- it delves a lot into mechanisms which are difficult to comment upon in a small study. 

c) Would suggest explaining more/removing the last paragraph on discussion of regression analysis 

Round 2

Reviewer 2 Report

The edited version looks much better. I have no further comments